# A tiny Triassic saurian from Connecticut and the early evolution of the diapsid feeding apparatus

Adam C. Pritchard[1], Jacques A. Gauthier[1,2], Michael Hanson[1], Gabriel S. Bever[3] & Bhart-Anjan S. Bhullar [1,2]

Following the Permo–Triassic Extinction, large-bodied diapsid reptiles—with a body length >1 m—rapidly expanded their ecological roles. This diversification is reflected in enormous disparity in the development of the rostrum and adductor chamber. However, it is unclear how marked the diversity of the feeding apparatus was in contemporary small-bodied diapsids. Here we describe the remarkably small skull (2.5 cm long) of a saurian reptile, *Colobops noviportensis*, gen. et sp. nov., from the Triassic New Haven Arkose of Connecticut, USA. The taxon possesses an exceptionally reinforced snout and strikingly expanded supratemporal fossae for adductor musculature relative to any known Mesozoic or Recent diapsid of similar size. Our phylogenetic analyses support *C. noviportensis* as an early diverging pan-archosaur. *Colobops noviportensis* reveals extraordinary disparity of the feeding apparatus in small-bodied early Mesozoic diapsids, and a suite of morphologies, functionally related to a powerful bite, unknown in any small-bodied diapsid.

[1] Department of Geology and Geophysics, Yale University, 210 Whitney Avenue, New Haven, CT 06511, USA. [2] Yale Peabody Museum of Natural History, Yale University, 170 Whitney Avenue, New Haven, CT 06511, USA. [3] Center for Functional Anatomy and Evolution, School of Medicine, Johns Hopkins University, 1830 E. Monument Street, Baltimore, MD 21205, USA. Correspondence and requests for materials should be addressed to A.C.P. (email: pritchardac@si.edu) or to B.-A.S.B. (email: bhart-anjan.bhullar@yale.edu)

The early Mesozoic Era saw the initial flowering of the major tetrapod clades that dominated terrestrial ecosystems thereafter. The earliest representatives of crown-group Amphibia, Lepidosauria, and Archosauria all appeared during the Triassic Period, such that the interval is now recognized as a key time in the emergence of major vertebrate clades[1,2]. However, in reviews of vertebrate history, there is a long-running assumption that these clades emerged from unspecialized, generalist ancestors during the Triassic[3]. However, this does not mean that the stem taxa from which those crown groups emerged did not produce substantial diversity of their own during the Triassic; indeed, recent studies of early pan-archosaur clades have revealed a substantial morphological radiation[4,5]. Unfortunately, much of our understanding of this record is biased towards large and medium-sized species (>1 m total body length).

In present-day ecosystems, significant morphological and ecological disparity, as well as a substantial proportion of taxonomic diversity, reside in small-bodied animals (<1 m total body length), whereas preservational biases often exclude such species from the fossil record[6]. These unfortunate biases likely masks a far greater ecomorphological disparity than has generally been appreciated among stem-members of major tetrapod clades radiating during the Permo–Triassic transition in the early Mesozoic. Indeed, recent discoveries of beautifully preserved small-bodied fossils in the mid-Mesozoic of China have refuted the classic hypothesis that the Mesozoic pan-mammals were homogenously small, shrew-like animals[7,8].

Here, we substantially expand the ecomorphological disparity of small-bodied vertebrates from the early Mesozoic with a new taxon from the Upper Triassic New Haven Arkose of Connecticut, USA[9]. The new form is represented by a very small partial skull (total skull length = 2.5 cm) of undetermined ontogenetic state first studied by ref. [9]. The new taxon displays specializations of the feeding apparatus unprecedented in any other known small tetrapod, juvenile or adult. Despite its small body size, the new species possesses hypertrophied attachments for the jaw adductor muscles that fill the transverse width of the postorbital portion of the cranium and a heavily reinforced rostrum with extensive overlap of adjacent bones. The supratemporal fossae in this species are proportionally broader than in any other comparably

sized crown-group reptile. Its presence at the dawn of the Mesozoic emphasizes that the major modern vertebrate clades originated in a world populated by small- and large-bodied ecomorphological extremes.

## Results

### Systematic palaeontology.

Sauria McCartney, 1802
Archosauromorpha von Huene, 1956
Rhynchosauria? Osborn, 1903b
*Colobops noviportensis* gen. et sp. nov.
"Sphenodontia indet." Sues & Baird, 1993

**Etymology.** From the Greek κολοβός (kolobós) for "docked" or "shortened" and ὤψ (ṓps) for "face". Refers to the abbreviated rostrum relative to most other known Triassic Diapsida. *Novus Portus* is a Latinized version of "New Haven", in reference to the discovery of the fossil in the New Haven Arkose in New Haven County, Connecticut, USA.

**Holotype.** YPM VPPU 18835 (Yale Peabody Museum, Vertebrate Paleontology Princeton Collection), nearly complete cranium and coronoid process of right mandible, lacking tooth-bearing portions of premaxillae, maxillae, palate, and mandible (Fig. 1).

**Locality and horizon.** (Based on ref. [9]) Near the junction of Routes 6A, 91, and 15 between Meriden and Middletown, CT, USA. The locality is part of the New Haven Arkose, Hartford Basin, Newark Supergroup. ref. [10] dated a section of the New Haven Arkose at 212 ± 2 Ma, indicating a middle Norian age.

**Diagnosis.** (1) Prominent, symmetrical fontanelle between frontals and parietals in midline (may reflect juvenility); (2) prefrontal and postfrontal contact one another to exclude frontal from orbit; (3) jugal maxillo-palatal junctions transversely broader than postorbital process of jugal; (4) dorsal exposure of postorbital transversely broad, with posteriorly directed process near the transverse midpoint of the supratemporal fenestra (may reflect juvenility); (5) palatine with transversely broad anterior concavity between lateral contact with maxilla and anteromedial contact with vomer.

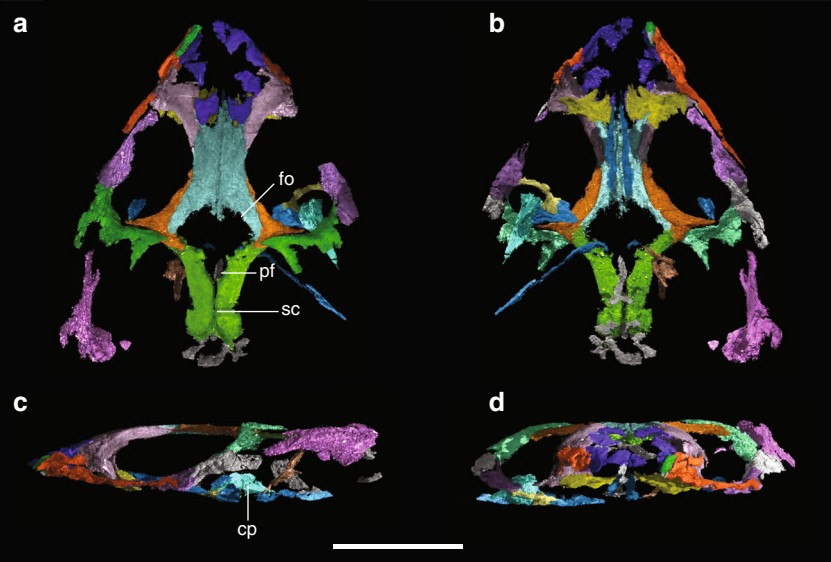

**Fig. 1** Reconstructed skull of *Colobops noviportensis*. Three-dimensional volume rendering of the skull of *Colobops noviportensis* (YPM VPPU 18835) produced in VG Studio Max 3.0 in **a** dorsal, **b** ventral, **c** left lateral, and **d** anterior views. Gray portions indicate portions of the skull of uncertain homology. Scale bar equal to 1 cm. Abbreviations: cp, coronoid process; fo, fontanelle; pf, parietal foramen; sc, sagittal crest

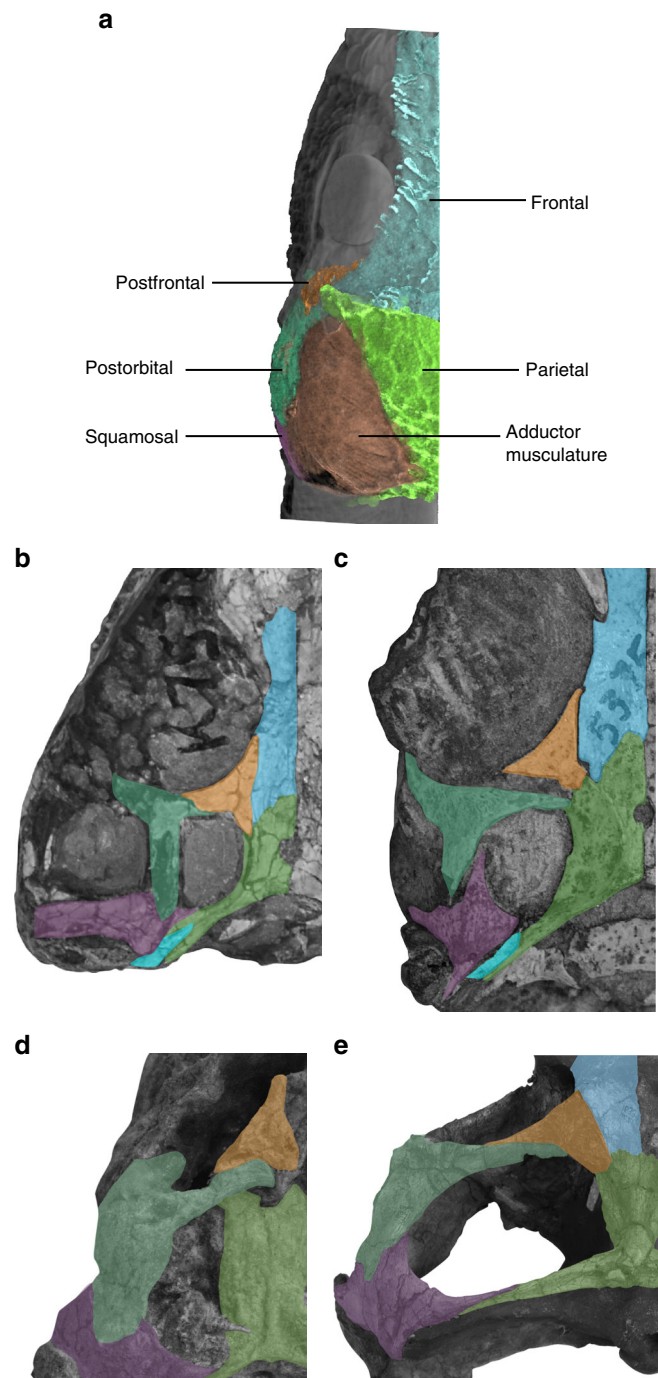

**Fig. 2** Supratemporal anatomy of Recent and Permo–Triassic Diapsida. Left orbital and supratemporal regions of a selection of diapsid reptiles in dorsal view. **a** Three-dimensional reconstruction of contrast-stained specimen of *Anolis sagrei*, illustrating the direct correlation between the osseus supratemporal fossa and the transverse breadth of the dorsal portion of the adductor muscle mass in diapsids. Photographs of the fossil skulls of **b** *Tropidostoma* Zone *Youngina* (SAM-PK 7573), **c** *Prolacerta broomi* (BP/1 5375), **d** juvenile hyperodapedontine rhynchosaur (MCZ 1664), and **e** *Hyperodapedon sanjuanensis* (MCZ 1636)

**Morphological description**. A series of measures of the *Colobops noviportensis* holotype is presented in Supplementary Table 1. The rostrum is extremely short proportionally, comprising less than a quarter of the overall length of the skull (Fig. 1a). The proportional height of the skull cannot fully be assessed, as the ventral (and dentigerous) portions of the premaxillae, maxillae, and pterygoids are not preserved. However, the configuration of the palatines and preserved portions of the pterygoid transverse processes suggests that the skull was depressed relative to its length (Fig. 1c). The enlarged orbits are proportionally long and strongly oriented dorsolaterally. A large, dorsally convex piece of bone sitting medial to the jugal is the right coronoid process and the only portion of the mandible preserved. Comparisons with other taxa, where not otherwise referenced, are based on specimens listed in Supplementary Notes 1 and 2.

The rostrum of *C. noviportensis* is heavily weathered and fragmented, although our µCT scan suggests that much of the dorsal portions of the nasals and maxillae are complete (Fig. 1a). We interpret a narrow splint of bone situated between the lateral margin of the left nasal and the medial margin of the left maxilla as the posterodorsal process of the premaxilla (Fig. 1a, d) based on comparisons with early diverging pan-archosaurs (e.g., *Mesosuchus browni*, *Prolacerta broomi*). The presence of this apomorphy supports the identification of *Colobops* as a pan-archosaur.

The antorbital region of *C. noviportensis* is exceptionally reinforced. The nasal exhibits a dorsoventrally tall ventrolateral lamina that braces the entire medial surface of the facial process of the maxilla (Fig. 1d), much as in Rhynchosauria (e.g., *M. browni*, *Teyumbaita sulcognathus*) and Rhynchocephalia (*Clevosaurus hudsoni*, *Sphenodon punctatus*). The prefrontal similarly braces roughly half the dorsoventral height of the facial process of the maxilla in testudinate turtles[11,12]. The palatine bears an unusually broad prefrontal process that meets the similarly broad orbital contribution of the prefrontal, forming a wide prefrontal pillar bracing the skull roof against the jaws, similar to the transversely broad contact in many lepidosaurs[13], turtles[11,12], and deeply nested Rhynchosauria[14,15]. At its juncture with the ectopterygoid, the central portion of the jugal is twice as transversely broad as the ascending (=postorbital process), similar to durophagous squamates such as *Adamisaurus magnidentatus*[16], *Chamaeleolis chamaeleonides*[17], and *Dracaena guianensis*[18].

The large orbits are directed somewhat dorsally, owing in part to the relatively low profile of the preserved skull, similar to modern *Sceloporus* spp. The prefrontal and postfrontal meet near the anteroposterior midpoint of the orbit, excluding the frontal entirely. The postfrontal is overall anteroposteriorly elongate, nearly excluding the medial process of the postorbital from contacting the parietal. The skull roof has a large, bilaterally symmetrical gap between the frontals and parietals in the midline. We identify this gap as a fontanelle based on the excellent preservation of the remainder of the skull roof and its symmetry on a sagittal plane. The posterior margin of the fontanelle is continuous with the transversely broad parietal foramen. The fontanelle may indicate that the specimen is a very young individual[18,19] although fontanelles are maintained to maturity in some iguanians[13,20].

The adductor chamber is remarkably voluminous in *C. noviportensis*; among sampled species—both Mesozoic and Recent (examples in Fig. 2). Indeed, based on a series of measures of early Mesozoic and Recent taxa, *C. noviportensis* is the smallest to possess such a proportionally broad space for jaw-closing musculature (Fig. 3). The supratemporal fenestra of *C. noviportensis* is proportionally broader than the orbital and preorbital portions of the skull (Fig. 1a). A similar condition arose

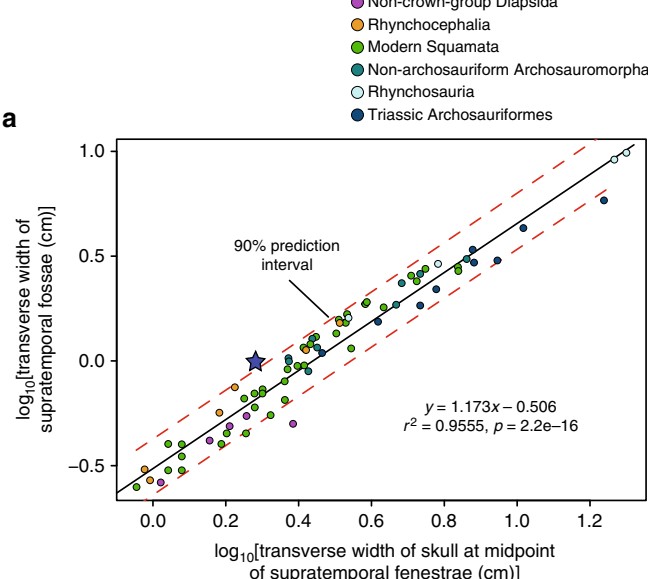

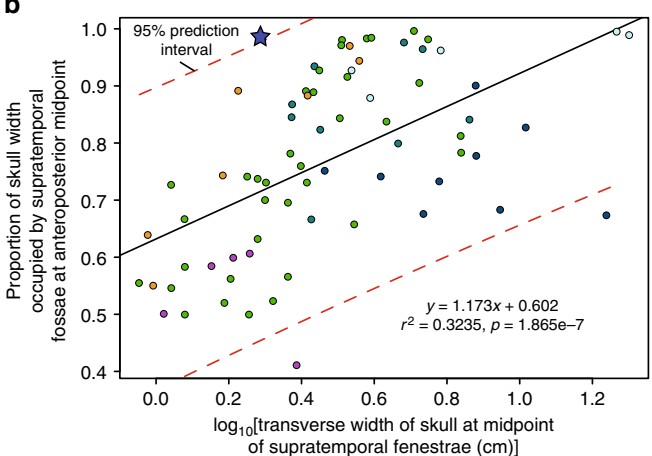

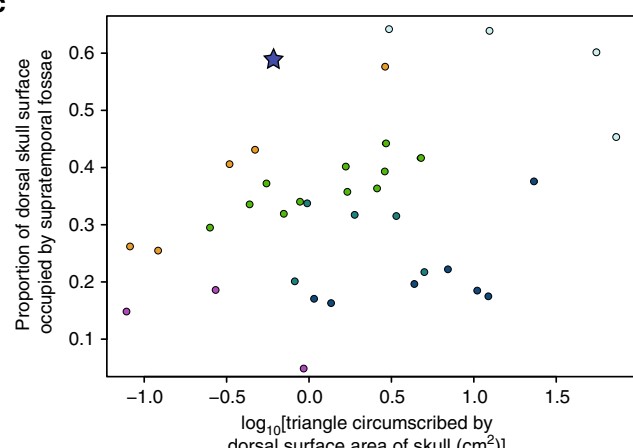

**Fig. 3** Plots of metrics of overall skull size against supratemporal fossae dimensions. **a** Bivariate plot of the log-transformed transverse width of the postorbital portion of the skull (measured at the anteroposterior midpoints of the supratemporal fenestrae) against the log-transformed transverse width of the supratemporal fossae at the equivalent point. **b** Bivariate plot of the log-transformed transverse width of the postorbital portion of the skull against the proportional contribution of the supratemporal fossae to the postorbital breadth of the skull. **c** Bivariate plot of the log-transformed dorsal surface area of the skull against the proportional contribution of the supratemporal fossae to the dorsal surface area of the skull. On all plots, the blue star indicates the measurements for *Colobops noviportensis*. Methods for data collection and statistical modeling are described in the Methods

Only small portions of the braincase and splanchnocranium are preserved (Fig. 1b). The supraoccipital has a narrow ascending process that fits between the posteromedial margins of the parietals and small dorsolateral processes that fit against the lateral edges of the supratemporal processes of the parietals. A similar bracing of supraoccipital against braincase occurs in captorhinomorphs[23,24]. The basioccipital is poorly preserved, represented by a horizontally oriented, flat plate of bone positioned ventral to the supraoccipital. In contrast to modern lepidosaurs, a probable ossified parasphenoid rostrum is preserved in *C. noviportensis* as a dorsoventrally tall, mediolaterally compressed plate of bone that sits on the midline anterior to the basioccipital. The left epipterygoid is massive, with a thick dorsal process oriented posterodorsally (Fig. 1b), indicating a well-developed m. pseudotemporalis profundus and further reinforcement of the braincase region.

The only portion of the mandible preserved is a large, dorsally convex right coronoid process. It is unclear which mandibular bones contribute to this structure. This element sits posterior to the ectopterygoid, medial to the jugal, and lateral to the pterygoid. The prominence of this portion of the mandible is comparable to some extant lepidosaurs (e.g., *Ctenosaura pectinata*, *Uromastyx hardwickii*) and to some early-diverging pan-archosaurs such as *Trilophosaurus buettneri*[25,26]. It is strikingly different from the coronoid eminence in other rhynchosaurs, which is formed by the posterodorsal slope of the dentary reaching the dorsally expanded coronoid[15].

**Characterization of adductor hyperdevelopment**. To characterize the relative size of the adductor chamber in *C. noviportensis*, we plotted the width of the dorsal exposure of the adductor muscle attachments relative to the total postorbital width of the skull in a range of extant and extinct diapsids. In the plot of skull size against adductor chamber width, *C. noviportensis* represents the smallest sampled diapsid to possess such a degree of adductor hypertrophy in terms of relative width and relative surface area (Fig. 3). The measures of *C. noviportensis* also fall above our calculated prediction intervals for these values. Measurements from all species used in this study are shown in Supplementary Table 1. Moreover, we identified a number of poorly understood trends in reptilian adductor evolution. In stem-Sauria, whether distantly related to the crown (e.g., the Pennsylvanian *Petrolacosaurus kansensis*) or closer to it (e.g., the Lopingian *Youngina capensis*, Fig. 2b), there is no embayment for the muscle mass on the lateral surface of the parietal and the space for the adductor chambers occupies <61% the total width of the postorbital skull. Among Permian and Triassic crown-group diapsids (=Sauria), early members of both lepidosaur and archosaur branches exhibit broader supratemporal fossae for the adductors, between 64% to nearly the full width of the skull in

independently in deeply nested Rhynchosauria[21,22]. There is a narrow, midline sagittal crest on the parietals posterior to the parietal foramen, indicating medial expansion of the m. adductor mandibulae profundus and m. pseudotemporalis superficialis. Proportionally, the adductor chamber occupies a similarly large percentage of the postorbital skull width as in hyperodapedontine rhynchosaurs and *Trilophosaurus buettneri*.

animals with very narrow sagittal crests (Fig. 2a, c–e). This has been noted as a phylogenetic character but never fully quantified[27–29] (percentage values obtained from dataset presented in Supplementary Table 1).

Within individual species, overall skull size appears to correlate strongly with the relative breadth of the adductor chamber; juveniles recapitulate the transition from Permian Diapsida to crown-group with a small supratemporal fossa with small proportionally modest embayments on the parietal giving way to proportionally larger fossae and deeper parietal embayments. Among modern Lepidosauria[30,31] and Archosauria[19,32], the dorsal attachments of the adductor musculature on the skull roof grow allometrically, becoming proportionally wider and more prominent relative to the breadth of the skull (example illustrated in Fig. 4) in contrast to other soft tissue structures of the head (e.g., eyes[33], brain[34,35]). Among sampled extant lepidosaurs, such crests occur in later ontogenetic stages of *Ctenosaura similis*, *Iguana iguana*, and *Sphenodon punctatus* (although we also note that size-related traits are sometimes decoupled from ontogeny in lepidosaurs[36]). Triassic stem-archosaurs with narrow sagittal crests include derived Rhynchosauria, *Trilophosaurus buettneri*, and *Tanystropheus longobardicus*. However, all sampled individuals with a sagittal crest—excluding *Colobops noviportensis*—have post-orbital skull widths >3 cm. If the *C. noviportensis* holotype represents an adult, the proportional size of its supratemporal fossae exceeds that in any other comparably sized diapsid. If it represents a juvenile, the proportional size of the fossae becomes more remarkable still, as similar-sized juveniles of Recent saurians have far smaller fossae.

**Phylogenetic analysis.** To investigate the affinities of *C. noviportensis*, we integrated the taxon into a phylogenetic analysis with broad sample of Permian and Triassic diapsids (modified from ref. [37]). Our phylogenetic analysis supports the hypothesis that *C. noviportensis* is a pan-archosaur and the earliest divergence within Rhynchosauria (*sensu* ref. [38]), linked by four unambiguous synapomorphies (Fig. 5). One is an anatomical character otherwise unknown in Triassic amniotes: (1) an anterior lamina of the maxilla that laps laterally over the posterior surface of the premaxilla (also present in *Mesosuchus browni* and *Rhynchosaurus articeps*). However, the analysis indicates that the clade represented by the Late Triassic *C. noviportensis* diverged shortly after the Permo–Triassic Extinction, as the oldest-known probable rhynchosaur, *Noteosuchus colletti*, dates to the Early Triassic[22]. However, this position is weakly supported; two additional steps produce topologies in which *C. noviportensis* occupies some positions with pan-Archosauria and a position nested within Sphenodontia, a clade that converged anatomically on rhynchosaurs in numerous skull characters (for exploration of alternative topologies, see Methods). Its exact phylogenetic position may be further obscured if the holotype indeed belongs to a juvenile—juvenile specimens incorporated into morphology-based phylogenies tend to root basal to adults of the same species[39,40]. However, known juvenile specimens of *Hyperodapedon*[41] already bear many synapomorphies that not only position them within Rhynchosauria, but also within Hyperodapedontinae (Supplementary Note 3). It is also possible that the branch including *C. noviportensis* and that including all other Rhynchosauria diverged rapidly following the divergence of Rhynchosauria as a whole, such that only a limited time existed

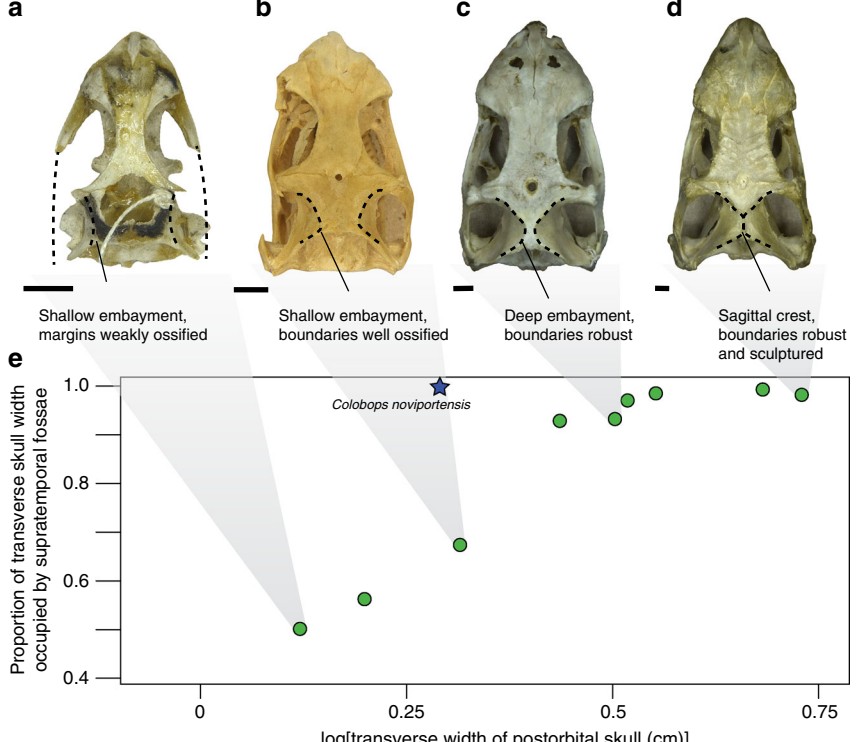

**Fig. 4** Plot of relative width of supratemporal fossae by total skull width in modern *Iguana*. Skulls of *Iguana iguana* in dorsal view, representing an ontogenetic series illustrating the broadening of the attachments of the adductor musculature (including **a** USNM VZ 220232, **b** USNM VZ 220231, **c** USNM VZ 220236, **d** USNM VZ 70470) and **e** bivariate plot of the transverse width of the postorbital portion of the skull (measured at the anteroposterior midpoints of the supratemporal fenestrae) against the proportional contribution of the supratemporal fossae to the transverse width of the postorbital portion of the skull using data from *Iguana iguana*. Shaded indicators indicate correspondence between data points and the photographs of the individual skulls. The star on the plot indicates the position of *Colobops noviportensis* (YPM VPPU 18835) in morphospace. Scale bars equal to 5 mm

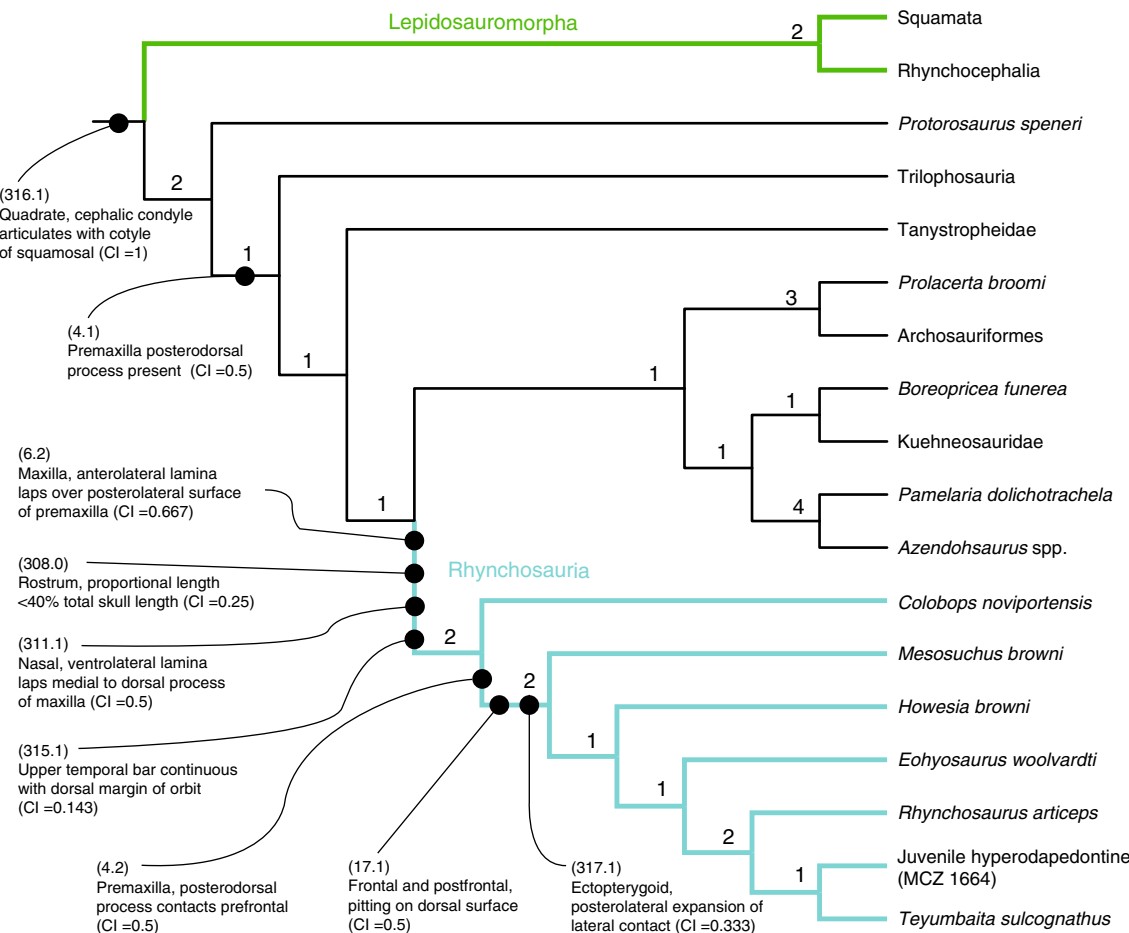

**Fig. 5** Phylogenetic affinities of *Colobops noviportensis*. Simplified strict consensus of most-parsimonious trees based on the species-level parsimony-based phylogenetic analysis of Permo–Triassic diapsids (CI = 0.326, RI = 0.644). Callouts indicate specific unambiguous synapomorphies preserved in YPM VPPU 18835 that support the phylogenetic hypothesis that *Colobops noviportensis* represents the earliest-diverging lineage in Rhynchosauria. The complete topology of the parsimony analysis is presented in Supplementary Fig. 13, whereas the topology from the Bayesian analysis is presented in Supplementary Fig. 14

for synapomorphies to accrue (suggested for early Diapsida[42]). Further study of early saurian phylogeny and more material of *C. noviportensis* are needed to resolve the species' phylogenetic heritage.

## Discussion

The taxonomic diversity of small-bodied reptiles in early Mesozoic terrestrial faunas has expanded rapidly in the past half century[26,43,44]. *Colobops noviportensis* expands this taxonomic diversity of early saurians. It also adds to our understanding of disparity in these taxa, displaying a suite of cranial apomorphies otherwise unknown in Triassic reptiles and indicating a specialized feeding strategy.

Although the *C. noviportensis* holotype lacks preserved teeth, the well-preserved skull roof enables inferences about the nature of its bite. In extant squamates, the highly differentiated jaw adductor musculature is usually accentuated in one of two ways[45,46]: (1) the parietals may be embayed medially by deep concavities for the m. pseudotemporalis superficialis and the m. adductor mandibulae externus profundus (as in *Ctenosaura pectinata*, *Iguana iguana*[47]); or (2) the postorbitals and squamosals may be expanded transversely to accommodate the m. adductor mandibulae externus superficialis (as in *Pogona vitticeps*, *Uromastyx hardwickii*). In *C. noviportensis*, both of these traits

are present, indicating that both portions of the adductor complex were enlarged.

The different roles the superficial and deep mandibular adductors play in food processing in extant saurians are not well understood. In a variety of extant squamates, electromyography studies show that all components of the mm. adductor mandibulae and mm. pseudotemporalis fire roughly simultaneously during nearly all portions of the feeding cycle[48,49]. The high coronoid process would have increased the moment arm of the attached jaw muscles, further enhancing the force of jaw closure[50]. Thus, for an animal of its body size, *C. noviportensis* likely had a more powerful bite than any other Triassic reptile.

Relatively large jaw muscle attachments in juveniles of modern Diapsida only occur in taxa that occupy ecomorphological extremes throughout ontogeny (e.g., the durophagous *Chamaeleolis chamaeleonides*[17]). Juvenile hyperodapedontine rhynchosaurs already possess a narrow sagittal crest[41], but the smallest known individuals are more than twice the size of the *C. noviportensis* holotype. The presence of a fontanelle in the *C. noviportensis* holotype gives us pause about arguing for its relative ontogenetic state, but the presence of such enlarged adductor chambers in such a small skull is unprecedented regardless.

The expanded space for jaw muscles occurs concomitantly with rostral reinforcement in *C. noviportensis*, suggesting a functional correlation. The large overlap of the nasal, maxilla, and

premaxilla differs from the comparatively little osseous contact between those elements in most early Sauria; indeed, only Rhynchocephalia and the much-larger Rhynchosauria approach this condition. Similarly broad overlap between nasal and maxilla also occurs in known Amphisbaenia (e.g., *Rhineura floridana*[13]; *R. hatcheri*[51]), highly modified head-first burrowers, some of which engage in durophagy[52,53]. No known taxon, extinct or extant, combines rostral and palatal expansion and reinforcement in the manner of *C. noviportensis*.

The best insights into the feeding of *C. noviportensis* come from the general shape of the adductor chamber. In *C. noviportensis*, the post-temporal process of the parietal is oriented laterally, as in Sphenodontia[13,54] and Rhynchosauridae[15,21], rather than posterolaterally as in most pan-lepidosaurs[13,55] and pan-archosaurs[2,38]. These former two taxa share precision bites, in which the dentary tooth row fits within a groove between multiple upper tooth rows; it is possible that the shape of the adductor chamber shared with *C. noviportensis* indicates a similarly complex dental anatomy. However, the actual use of this highly specialized grooved tooth row likely differed greatly between these taxa. *Sphenodon punctatus* employs a distinctive, propalinal motion in which the lower tooth row slides mesiodistally between palatal and upper tooth rows[56,57], whereas rhynchosaurids likely used a precision-shear bite[21,58]. These comparisons may indicate a similarly complex dental apparatus in *C. noviportensis*, but they are equivocal about the precise nature of the bite.

The increasing taxonomic and ecological diversity of Diapsida throughout the Permo–Triassic transition occurred in concert with a transition from relatively simple, unspecialized adductor chambers to a variety of distinct morphologies. The hypertrophied jaw muscles and reinforced rostrum at the small size of *C. noviportensis* illustrate that the extremes of feeding diversity did not solely involve large-bodied animals. No other known early Mesozoic taxon of similar body size possessed the combination of a powerful biting apparatus and a reinforced bony rostrum.

Reconstructing evolutionary radiations is a key goal of historical biological research. Many recent studies on the morphological histories of small-bodied vertebrate groups have incorporated only samples of extant species[59,60]. However, any attempt to conceptualize morphological radiations in deep time must be accompanied by continued sampling of the fossil record. *Colobops noviportensis* represents a combination of morphological traits unknown in extant amniotes, and thus a morphology that would not have been reconstructed in a macroevolutionary analysis based exclusively on extant species. The fossil record, when rigorously sampled and analyzed at all scales, has the potential to expand our understanding of the limits of morphology and ecology.

## Methods

**µCT scanning and three-dimensional reconstruction**. YPM VPPU 18835 was µCT scanned at the Laboratory of Integrative Science's Center for Nanoscale Systems at Harvard University (Cambridge, MA) by A. Pritchard. The specimen was initially scanned at a resolution of 0.01566797 mm for a total of 1812 slices (60 kV, 285 mA). There is a strong contrast between bone and matrix, although the sandstone does contain some similarly radio-opaque clasts. A volume rendering of the µCT data prior to segmentation is presented in Supplementary Fig. 1.

The individual bones of the specimen were segmented in VG Studio Max 3.0. We filtered these out of the resultant segmented bones by identifying textural differences between the dense bone and diffuse sandstone inclusions. Sutural boundaries were easy to identify in the palate and post-orbital regions. Narrow gaps were identified in the rostrum that demarcated the sutures between the nasals and maxillae. The identification of the premaxilla is based on (1) the separation between this element and the overlying maxilla and (2) the clear demarcation of two separate contacts for lateral elements on the lateral surface of the nasals. We were unable to identify a clear contact between the postorbital and jugal on either

side. The central portions of the postorbital bars on both sides have been rendered in gray to emphasize the ambiguity of the reconstruction in these areas.

The specimen of *Anolis sagrei* reconstructed in Fig. 2a was stained using the contrast-staining agent phosphomolybdic acid. The specimen was µCT scanned at the Laboratory of Integrative Science's Center for Nanoscale Systems at Harvard University (Cambridge, MA, USA) by B.-A. Bhullar. The specimen was initially scanned at a resolution of 0.0206 mm for a total of 1963 slices (80 kV, 85 mA) using a molybdenum target and a one-second exposure.

The scan data of *Mesosuchus browni* (SAM-PK 6536) used for this phylogenetic analysis was µCT scanned at the University of Texas Austin High-Resolution X-ray Facility by Matt Colbert (Austin, TX, USA). The specimen was scanned at a resolution of 0.07837 mm for a total of 1122 slices (200 kV, 180 mA).

We present a series of supplemental movies in .avi format to accompany the anatomical description of *Colobops noviportensis* presented in the main text. These include the three-dimensional reconstruction of *Colobops noviportensis* presented in Fig. 1 (Supplementary Movie 1), animations of the original slice data from our CT scan (Supplementary Movies 2–4), and animated cutaways of the three-dimensional volume rendering of the holotype (Supplementary Movies 5–7). These are also uploaded to Morphobank Project 2649. We also present several CT slices from a scan of *Mesosuchus browni* (SAM-PK 6536).

**Measurements of supratemporal fossa and skull width**. To assess the relative contribution of the muscle-bearing supratemporal fossa to the postorbital skull width, we measured the transverse width of the fossae at the anteroposterior midpoint of the supratemporal fenestra in a sample of extant and extinct reptiles (sample listed in Supplementary Notes 1 and 2). We inferred that the supratemporal fossae demarcate the limits and attachments of the jaw adductor musculature in crown-group Reptilia based on the conditions in Sauria (e.g.,[45,46]) a Level I inference per[61]. The same inference on stem-group Reptilia (e.g., *Youngina capensis*) is based on anatomical correlates, a Level II inference. The Recent sample was taken from the Yale Peabody Museum vertebrate zoology collections and measured using millimeter-scale Beads Landing calipers. The fossil sample and those from the National Museum of Natural History collections were measured using a sample of photographs and images in literature of skulls in dorsal view using the measure function in Fiji[62]. These values were plotted using the "plot" function in R v. 3.3.3[63]. The resulting data points were color-coded and the axes labeled in Adobe Illustrator CC 21.0.2.

The full list of measurements presented in Fig. 3 is presented in Supplementary Table 1. We present a number of individual plots of lepidosaur taxa for which we measured a size-variable series to illustrate size-linked trends in the proportional size of the adductor chamber and supratemporal fossae. We first measured the breadth of the entire postorbital portion of the skull at the anteroposterior midpoint of the supratemporal fenestrae. At the same anteroposterior point, we measured the width of the supratemporal fossae. The proportional measures plotted indicate the relative contribution of the supratemporal fossae to the total postorbital breadth of the skull. The same data points are presented in a plot in which the *x*-axis is not log-transformed are presented in Supplementary Fig. 2. Plots, in which the *x*-axis is log-scaled, are presented for the lepidosaurs *Sphenodon punctatus* (Supplementary Fig. 3), *Iguana iguana* (Supplementary Fig. 4), *Pogona vitticeps* (Supplementary Fig. 5), *Sceloporus occidentalis* (Supplementary Fig. 6), *Teius teyou* (Supplementary Fig. 7), *Physignathus lesuerrii* (Supplementary Fig. 8), and *Hydrosaurus pustulatus* (Supplementary Fig. 9). For Supplementary Fig. 9, an ontogenetic series of skulls of the taxon is presented in dorsal view.

For the subsample of specimens for which the complete or near-complete anteroposterior length of the skull could be measured from scaled photographs, we obtained measures of the total anteroposterior length of the skull from the anterior most tips of the preserved premaxillae. Specimens were excluded when neither the premaxillae nor the margins of the external nares were preserved. We generated a measure of the triangle circumscribed by this length measurement and the transverse width measurement to generate a proxy of the dorsal surface area of the skull (log-transformed and used for the *x*-axis in Fig. 3c). For each of these skulls, we measured the surface area of the skull roof occupied by the supratemporal fossa from photographs by drawing the boundaries of the opening. The length, width, and supratemporal fossa area measurements were obtained using scaled photographs in Fiji v. 1.0 for OSX[62].

**Statistics for skull proportions measures**. We examined the relationship between (a) the transverse width of the skull and the transverse width of the supratemporal fossae and (b) the transverse width of the skull and the proportional width of the supratemporal fossae using the regression modeling functions in R. The regression models resultant from these analyses and their $r^2$ values are presented in Fig. 3. For both of these linear models, we also present prediction intervals developed using the predict function in R.

To examine the statistical relationships between measures of skull size and supratemporal fossa size (both absolute and proportional), we generated linear models for the relationships in R[63]. For absolute measurement variables that we plotted and analyzed (transverse breadth of the skull, transverse breadth of the supratemporal fossa, dorsal surface area of the skull), we log-transformed all measurements. We did not transform our proportion measurements (relative width of the supratemporal fossa, relative surface area of the supratemporal fossae),

as they did not exhibit an evident binomial distribution. We employ skull width as the descriptive variable for overall skull size rather than anteroposterior length because of the large number of specimens in our dataset that lack a complete rostrum but possess well-preserved skull tables.

We developed linear models in R for the two relationships that exhibited a correlation: (a) the log-transformed values of skull width and supratemporal fossa width and (b) the log-transformed value of skull width and the proportional contribution of the supratemporal fossa to the width of the skull. The resultant equations and $r^2$ values are presented in Fig. 3. In both of these cases, the measurements for *C. noviportensis* (YPM PU 18835) were excluded from the datasets in order to assess how far the taxon diverged from estimates. We used the log-transformation for raw measurements on the recommendation of ref. [64]. However, we did not log-transform the proportion values, again on the recommendation of ref. [64].

For both of these two relationships, we also used the R function for generating prediction intervals. In the case of the absolute measures, the measures for *C. noviportensis* fall above the maximum-predicted value for the 90% prediction interval. In the case of the model describing the relationship between absolute skull width and the proportional width of the supratemporal fossae, the relative proportions of the supratemporal fossae in *C. noviportensis* fall above the maximum-predicted value for the 95% prediction interval. For the plot illustrating the proportional contribution of the surface area of the supratemporal fossae to the skull roof, *C. noviportensis* possesses proportionally larger supratemporal fossae than any other taxon measured of an equivalent skull size.

**Phylogenetic analysis**. We analyzed the affinities of *Colobops noviportensis* using a phylogenetic dataset focused on Permo–Triassic Diapsida. The data matrix used is a modification of ref. [37], itself a combination of refs. [43,65,66]. A full listing of material and phylogenetic characters used for the phylogenetic analysis is presented in Supplementary Notes 1 and 2. Modifications to the dataset are presented in Supplementary Notes 3 and 4. The dataset incorporates 57 terminal taxa and 323 characters. Illustrations of key morphological character states are provided in Supplementary Figs. 10–13.

Our parsimony analysis was run in TNT v. 1.1[67], employing the "Traditional Search" options including 10,000 replicates of Wagner trees (using random addition sequences), followed by tree bisection and reconnection (TBR) holding 10 trees per replicate. The best trees obtained at the end of the replicates were subjected to a final round of TBR branch swapping. We employed Rule 1 of ref. [68] for collapsing zero-length branches. We recovered 2 most-parsimonious trees of 1,100 steps. We employed the STATS.RUN TNT script to obtain the Consistency Index (CI = 0.326) and Retention Index (RI = 0.644) for all trees. We used the Bremer support option in the Trees submenu of TNT, calculating supports based on a new round of TBR, holding trees suboptimal by 15 steps. The most-parsimonious trees infer *Colobops noviportensis* as the sister taxon of *Mesosuchus browni* + all other Rhynchosauria. This clade has a Bremer support of 2, but it is not recovered in Bootstrap resampling analyses. We explored alternative phylogenetic positions for *C. noviportensis*, finding a position among Sphenodontia suboptimal but only by two steps (see Methods). The strict consensus is provided in Supplementary Fig. 14.

The topology is otherwise largely consistent with the parsimony analysis of ref. [37]. However, this parsimony analysis does not infer a monophyletic Allokotosauria (defined in ref. [66] as "[t]he least-inclusive clade containing *Azendohsaurus madagaskarensis*, Flynn et al., 2010, and *Trilophosaurus buettneri*, Case, 1928a, but not *Tanystropheus longobardicus*, Bassani, 1886, *Proterosuchus fergusi*, Broom, 1903, *Protorosaurus speneri* von Meyer, 1830, or *Rhynchosaurus articeps* Owen, 1842"). Instead, Trilophosauria is the second divergence within Archosauromorpha following *Protorosaurus speneri*. Azendohsauridae (including *Pamelaria dolichotrachela*) is the sister taxon of a *Boreopricea funerea* + Kuehneosauridae clade.

A Bayesian analysis was run in MrBayes v. 3.2[69], using the Mk model using a gamma distribution for rate variation across sites. The analysis was performed with a sampling frequency of 1000, two concurrent runs (nruns = 2), and four Metropolis-coupled chains (nchains = 4) for 20,000,000 generations. A relative burn-in of 25% was used. The Potential Scale Reduction Factor (PSRF+) hovered around 1.0 as the runs converged, and the effective sample size (ESS), examined in Tracer v. 1.6[70] were substantially >200. The output from MrBayes following the 20,000,000 generations is uploaded to the 'Documents' section on Morphobank Project 2816. The results of the Bayesian analysis are presented in Supplementary Fig. 15. As in the parsimony analysis, the Bayesian analysis infers *Colobops noviportensis* as the sister taxon of *Mesosuchus browni* + all other Rhynchosauria. The clade is not particularly well supported, with a posterior probability of 0.6373.

The topology is consistent in most respects with the strict consensus of the most-parsimonious trees, with a few exceptions. The Bayesian analysis recovered Weigeltisauridae as the earliest-diverging diapsid clade after *Orovenator mayorum*, rather than Drepanosauromorpha as in the parsimony analysis. The Malagasy Permian diapsids are also not recovered in a single clade, rather occurring in a polytomy with *Youngina* and Sauria. *Macrocnemus* is not monophyletic; instead, *M. bassanii* and *M. fuyuanensis* form successive sisters to the rest of Tanystropheidae.

Intriguingly, the Bayesian analysis recovers a topology consistent with the parsimony analysis of ref. [37]: Kuehneosauridae, *Boreopricea funerea*, and

Trilophosauria form a single clade within Archosauromorpha. This result contrasts with the strict consensus recovered in the current parsimony analysis, in which a Kuehneosauridae + *Boreopricea funerea* clade is sister to Azendohsauridae. This clade is quite unstable; it is not recovered in the Bootstrap analysis and it has only a Bremer support of 1. The posterior probability of the (Kuehneosauridae + *Boreopricea funerea*) + Trilophosauria in the Bayesian analysis is 0.7108.

**Constraint analysis**. The results of our primary phylogenetic analyses are presented in Supplementary Figs. 14 (for the parsimony analysis) and 15 (for the Bayesian analysis). The limited support values for the phylogenetic position of *Colobops noviportensis* among Diapsida suggests that alternative phylogenetic hypotheses may be supported by small amounts of additional data. We chose to explore the pre-existing hypothesis that the *C. noviportensis* (YPM VPPU 18835) represents a sphenodontian rather than a pan-archosaur as originally suggested by ref. [9]. To do so, we constrained *C. noviportensis* to be in a clade with the other Sphenodontia integrated into this analysis (*Diphydontosaurus avonis*, *Planocephalosaurus robinsonae*, *Clevosaurus hudsoni*, and *Clevosaurus brasiliensis*). The analysis with this constraint enforced was run in TNT v. 1.5 with identical parameters to the main parsimony analysis. The strict consensus of the minimum length trees from this constraint study is presented in Supplementary Fig. 16.

The resulting constraint analysis produced 13 minimum-length trees of 1102 steps (hit 6113 out of 10,000 replicates), with a CI = 0.326 and RI = 0.643. This value is only two steps longer than the most-parsimonious trees reported. In these suboptimal trees, *Colobops noviportensis* is resolved as the sister taxon of *Clevosaurus brasiliensis* + *Clevosaurus hudsoni*. This position of *C. noviportensis* is not particularly surprising owing to the suite of morphological characters shared between *Clevosaurus*, derived sphenodontians, and derived pan-archosaurs in this analysis. In these sub-optimal trees, *Clevosaurus* independently acquired a posterodorsal process of the premaxilla excluding the maxilla from the posterior margin of the naris. Other synapomorphies of the clade of *Colobops noviportensis* + *Clevosaurus* recovered by this analysis include: 14:0, frontals unfused to one another in midline; 311:1, ventral lamina of nasal fitting against the full dorsoventral height of dorsal process of maxilla; 315:1, dorsal margin of postorbital bar at same dorsoventral level as dorsal margin of orbit. Each of these represent convergences between derived Sphenodontia and pan-archosaurs.

Frustratingly, the only preserved portions of the holotype skull of *Colobops noviportensis* includes structures that evolved convergently in derived Rhynchocephalia and Rhynchosauria. Further discoveries, especially elements of the postcranium would be extremely useful in elucidating the affinities of the species. *C. noviportensis*, however, does possess a number of character states unknown in Sphenodontia, including: 252(0), lacrimal present as distinct ossification; 313(0), postorbital medial process sits posteroventral to postfrontal. The current character data available for *C. noviportensis* supports its affinity with Rhynchosauria, but we recognize the limited support for that hypothesis and the possibility that revisions to phylogenetic datasets and new discoveries of *C. noviportensis* will alter the topology.

**Nomenclatural acts**. This published work and the nomenclatural acts it contains have been registered in ZooBank, the proposed online registration system for the International Code of Zoological Nomenclature (ICZN). The ZooBank LSIDs (Life Science Identifiers) can be resolved and the associated information viewed through any standard web browser by appending the LSID to the prefix "http://zoobank.org/". The LSIDs for this publication are: zoobank.org:act:A1500DA4-6D40-43FF-A771-6C3EEB1DE295 and zoobank.org:act:3C89EFD7-4E31-4C41-BBE3-7141BCE42A72.

**Data availability**. The phylogenetic data matrix, inputs and outputs for the MrBayes analysis, the input and linear models for R, and supplemental movies of CT data and three-dimensional volume renderings are available on Morphobank (www.morphobank.org) Project 2816. Original CT data are available from the corresponding authors upon request.

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

## Acknowledgements

We are grateful to R.G. Conrod for originally discovering the specimen in Meriden, CT in 1965 and donating the specimen to the Yale Peabody Museum. Marilyn Fox (YPM VP) further prepared the specimen after the prior publication by ref. [9]. We thank C. Mehling and M. Norell (AMNH); A. Henrici (CM); S. Chapman, L. Steel, and P. Barrett (NHMUK); K. de Queiroz and A. Wynn (USNM VZ); J. Cundiff (MCZ); and G. Watkins-Colwell (YPM VZ) for access to curated museum specimens. A. Pritchard is funded by the National Science Foundation (DEB 1501851, BIO 1523871). J.A.G. and B.-A.S.B. received partial financial support from the Yale Peabody Museum of Natural History.

## Author contributions

A.C.P., J.A.G., and B.-A.S.B. conceived the project. A.C.P. and G.S.B. collected and analyzed the data. A.C.P., J.A.G., M.H., G.S.B., and B.-A.S.B. designed the figures and wrote the paper.

## Additional information

**Competing interests:** The authors declare no competing interests.

