## [Peer Review File(PDF 423 kb) · Nature Communications]

Reviewers' comments:

Reviewer #1 (Remarks to the Author):

Revision of "Anatomical evolution of the diapsid feeding apparatus revealed by a tiny saurian from the Triassic of Connecticut" by Pritchard and colleagues.

In this manuscript it is redescribed a small diapsid skull from the Late Triassic of the USA that was previously interpreted as an indeterminate sphenodontian by Sues and Baird (1993). The authors provide substantial new anatomical information based on μ CT scan data and a volume rendering and based on this specimen proposed a new genus and species. The authors found this new taxon as the earliest branching rhynchosaur based on quantitative phylogenetic analyses and propose that its morphology considerably expands the ecomorphological disparity of early Mesozoic tetrapods.

The interpretations obtained from the μ CT scan data seem to be well-supported and the description is very informative. The figures are very informative, clear, and extremely well complemented with the videos of the supplementary information. The text is clear and well written –I have suggested only minor modifications listed at the bottom of this letter.

The results of the phylogenetic analyses are very interesting and have important implications in the evolutionary history of archosauromorphs, mainly that of rhynchosaur. The position of the new taxon as the earliest branching rhynchosaur results in a long ghost lineage of approximately 40 million years and in a morphology at the base of the clade that was rather unexpected. Thus, because of the latter implications, the position of the new taxon should be rather well supported. I checked the four synapomorphies that the authors found for Rhynchosauria (i.e. that support the position of the new taxon within Rhynchosauria) and I have the following queries:

1) The authors scored the presence a maxilla overlapping the premaxilla for Mesosuchus (character state 6.2). However, after checking some photographs and first hand observations, I consider that the postnasal process of the premaxilla overlaps the maxilla on the right side of the skull of SAM-PK-6536 (see attached photograph in the Word version of this file). I may be misinterpreting the character-state, but I think that the authors should double check the scoring of this character for Mesosuchus.

2) The authors claimed in several parts of the text that the holotype of Colobops may be a juvenile (e.g. presence of a fontanelle on the skull roof). Thus, the extremely short rostrum (character-state 308.0) may be a result of the juvenile stage of the specimen. I would not score this character for juvenile or possible juvenile (as it is the case in Colobops) specimens.

3) Character 311.1: can you show somewhere (e.g. supplementary information) the presence of a nasal with a ventrolateral lamina lapping medial to the dorsal process of the maxilla in SAM-PK-6536 (Mesosuchus browni)? I am struggling to figure out the condition based on the exposed surface of the snout of this specimen.

4) Character-state 315.1: based on the position of the anterior process of the squamosal in the volume rendering of the skull of Colobops, it seems that the upper temporal bar is placed approximately at mid-height of the orbit. Please, can you justify and maybe figure why do you consider that the upper temporal bar is placed at level with the dorsal border of the orbit?

Thus, based on the comments raised above, I am a bit sceptical about the phylogenetic placement of Colobops. I strongly encourage the authors to justify better the scorings of these character-states and respond to the queries listed above expanding the supplementary information. It would be very useful if the authors can illustrate these character-states for key taxa (e.g. some conditions in Mesosuchus).

Beyond the phylogenetic placement of the new taxon, the authors found the enigmatic kuehneosaurids as deeply nested within Archosauromorpha, a result that strongly expands the ecomorphological disparity of the group during the Triassic. The relevance of this result competes with the importance of Colobops for the early evolutionary history of saurians. However, this result is not justified at all in the main text, neither in the supplementary information. I the position of kuehneosaurids as deeply nested within Archosauormorpha should be discussed and clearly justified in this manuscript.

I consider that this manuscript is interesting and worth of publication, but the authors should successfully address the points raised above in the revised version of the manuscript. Thus, I recommend major changes for this manuscript.

Minor comments (main text)

Lines 62–64: "Its presence at the dawn of the Mesozoic indicates that the major modern vertebrate clades originated in a world populated by small- and large-bodied ecomorphological extremes."

I think that the relevance of Colobops for this statement is inflated. There are multiple small-bodied tetrapod taxa during the early Mesozoic that considerably increases the ecomorphological disparity of the clade, such as procolophonids and early lepidosauromorphs. I would suggest rewording the sentence in a way that the presence of the new taxon increases or enriches the ecomorphological disparity, but not indicates by itself the presence of such ecomorphological extremes.

Line 96: delete ", ".

Line 190: delete "small".

Line 210: "represented" is misspelled.

Line 313: check if "mA" would be changed to "μA".

Line 328: delete repeated "was".

Line 336: "midpoint" is misspelled.

Line 348, Phylogenetic analysis: can you add in this section how many characters and terminals has the modified version of the data matrix?

Line 353, Phylogenetic analysis: can you add how many trees have you recovered after the search and how many times the best score was recovered in the replicates?

Reference 55: "*Azendohsaurus madagaskarensis*" has to be in italics.

Figure caption 1: indicate that the central bars of the postorbital on both sides are rendered in grey to show ambiguity in the reconstruction.

Figure caption 2: "*Scaphonyx fischeri*" is currently considered a nomen dubium (see Langer and Schultz, 2000). As a result, I suggest identify this specimen as *Hyperodapedon* sp.

Figure caption 2: MCZ 1636 is referred by Sill (1970) to "*Scaphonyx fischeri*". Thus, can you justify somewhere in the text why you are referring this specimen to *Hyperodapedon sanjuanensis*?

Figure caption 4: "species-level" is misspelled.

Figure 1: there are some white lines in the black background that should be deleted.

Reviewer #2 (Remarks to the Author):

This is an important announcement of a new Triassic reptile, and well written and illustrated. Its importance is that the new species is sister to the rhynchosaurs, a major clade of herbivores, and offers a very new insight into diversity and adaptation. The focus on a small, rather than, large animal is also of interest. This is a notably weird little critter, showing many rhynchosaur-like features (reinforced snout, unusually broad adductor chamber), and yet, as sister to Rhynchosauria, enforcing an unexpected 30-million-year long ghost range.

Methodologically, this is a nice paper because the rather tatty specimen has been CT scanned, so revealing important details inside the block which would otherwise not have been visible.

Can the authors be 100% confident about the great posterior width of the skull/ size of adductor chamber in that this is demarcated only on the left-hand side, and it depends entirely on the position chosen for the squamosal (?) – the purple bone with bifurcated

anterior process – that is adrift from the other skull bones.

192: saggittal = sagittal

210: represneted = represented

Reviewer #3 (Remarks to the Author):

This is a very interesting paper on a small reptile from the late Triassic Newark Supergroup. Small animals are really important because they often demonstrate evolutionary novelties, and are underrepresented in the fossil record.

The authors describe a small saurian with possible affinities to archosauromorphs. This small specimen seems to preserve dramatically expanded space for the jaw adductor musculature (although sadly the lower jaws are absent), which has implications for feeding adaptations of the animal. This is all really interesting and worth publishing.

However, it is the quantification for this claim that I think requires a little more work on the part of the authors. When something that is extremely small also has an "unusually large" anything, one should immediately suspect an allometric affect, as being small magnifies the relative size of structures. No study of allometry was presented by the authors, so the reader has no means for judging whether the jaw musculature architecture is absolutely larger or not.

The authors present their Figure 3 in which the proof for the large size is presented. Their argument is that their specimen is "a clear outlier". I am uncertain that it is clear--in fact it looks like it is within the cloud of the distributed data presented. Looking at some of the other bivariate plots presented in the supplemental information, individual species exhibit possible non linear trends (where sufficient data are known). Axes should therefore be log transformed. I suspect when this is done that outlier would be brought into the total range of variation--if not then its "outlierness" can be tested via residual analysis, or discriminant analysis (especially when size can be removed as a variable).

The rest of the paper is fine and very interesting. The figures are all good, although the palatal morphology is really small and hard to see clearly. I wish there was more braincase material! The phylogenetic analysis is based on a data set that is well vetted from the literature; characters all appear independent and the new characters well defined, as to be expected from this research group. There is a typo in Figure 4 caption. General support metrics for the presented strict consensus tree would be helpful to be included.

REVIEWER 1 –

“Revision of “Anatomical evolution of the diapsid feeding apparatus revealed by a tiny saurian from the Triassic of Connecticut” by Pritchard and colleagues.

In this manuscript it is redescribed a small diapsid skull from the Late Triassic of the USA that was previously interpreted as an indeterminate sphenodontian by Sues and Baird (1993). The authors provide substantial new anatomical information based on μ CT scan data and a volume rendering and based on this specimen proposed a new genus and species. The authors found this new taxon as the earliest branching rhynchosaur based on quantitative phylogenetic analyses and propose that its morphology considerably expands the ecomorphological disparity of early Mesozoic tetrapods.

The interpretations obtained from the μ CT scan data seem to be well-supported and the description is very informative. The figures are very informative, clear, and extremely well complemented with the videos of the supplementary information. The text is clear and well written –I have suggested only minor medications listed at the bottom of this letter.

The results of the phylogenetic analyses are very interesting and have important implications in the evolutionary history of archosauromorphs, mainly that of rhynchosaurs. The position of the new taxon as the earliest branching rhynchosaur results in a long ghost lineage of approximately 40 million years and in a morphology at the base of the clade that was rather unexpected. Thus, because of the latter implications, the position of the new taxon should be rather well supported. I checked the four synapomorphies that the authors found for Rhynchosauria (i.e. that support the position of the new taxon within Rhynchosauria) and I have the following queries:

1) The authors scored the presence a maxilla overlapping the premaxilla for *Mesosuchus* (character state 6.2). However, after checking some photographs and first hand observations, I consider that the postnarial process of the premaxilla overlaps the maxilla on the right side of the skull of SAM-PK-6536 (see attached photograph). I may be misinterpreting the character-state, but I think that the authors should double check the scoring of this character for *Mesosuchus*.”

Contact between the premaxilla and maxilla.

- We have included a number of additional figures and CT data of the best-preserved skull of *Mesosuchus browni* (SAM PK-6536) as a justification for this coding. The coding was initially based on the illustrations by Dilkes (1998) of SAM PK-5882, which seemed to show a depression on the posterolateral surface of the premaxilla for the contact with the maxilla, similar to the lapping contact evident in *Rhynchosaurus articeps* (NHMUK R 1237) and *Hyperodapedon sanjuanensis* (MCZ 1636). However, we concur with Reviewer 1 that the contact in *Mesosuchus browni* (SAM PK-6536) does not show this sort of contact superficially.
- Our μ CT scan data of SAM PK-6536 shows a small anterolateral lamina of the maxilla lapping over the lateral surface of the posterodorsal process of the premaxilla, such the latter is “wedged” against the anterior surface of the maxilla. This stands in contrast to other pan-archosaurs, in which the premaxilla fits across the anterolateral surface of the maxilla. These states are illustrated in the new supplemental figures 9 and 10.

“2) The authors claimed in several parts of the text that the holotype of *Colobops* may be a juvenile (e.g. presence of a fontanelle on the skull roof). Thus, the extremely short rostrum (character-state 308.0) may be a result of the juvenile stage of the specimen. I would not score this character for juvenile or possible juvenile (as it is the case in *Colobops*) specimens.”

Proportions of the rostrum

- We concur that relative rostral length could be substantially influenced by ontogeny. We ran a permutation of the analysis in which this character was coded as “?” for *Colobops noviportensis*. Tree length and topology were not altered in this permutation.

“3) Character 311.1: can you show somewhere (e.g. supplementary information) the presence of a nasal with a ventrolateral lamina lapping medial to the dorsal process of the maxilla in SAM-PK-6536 (*Mesosuchus browni*)? I am struggling to figure out the condition based on the exposed surface of the snout of this specimen.”

Anterolateral lamina of maxilla

- Our initial coding of this condition in *Mesosuchus browni* was based on author A. Pritchard's interpretation of the internal surface of the rostrum during first hand study of the specimen. However, he was unable to obtain corroborating images due to the internal position of this structure. However, the prominence and depth of this lamina is apparent in the CT scan data and three-dimensional reconstructions of the specimen (See Figs. S10, S11).

"4) Character-state 315.1: based on the position of the anterior process of the squamosal in the volume rendering of the skull of *Colobops*, it seems that the upper temporal bar is placed approximately at mid-height of the orbit. Please, can you justify and maybe figure why do you consider that the upper temporal bar is placed at level with the dorsal border of the orbit?"

Height and position of postorbital bar.

- I believe Reviewer 1 is interpreting the gray bony bar illustrated on the segmented, volume-rendered skull as the base of the posterior process of the postorbital. Although we are not certain whether or not this portion of bone belongs to the jugal or postorbital, there is a substantial, posteriorly broken margin further dorsally on the definitive postorbital that is closely aligned to the clear facet on the anterior process of the squamosal.

"Thus, based on the comments raised above, I am a bit sceptical about the phylogenetic placement of *Colobops*. I strongly encourage the authors to justify better the scorings of these character-states and respond to the queries listed above expanding the supplementary information. It would be very useful if the authors can illustrate these character-states for key taxa (e.g. some conditions in *Mesosuchus*)."

- We concur with Reviewer 1 that illustrations are key to justifying these codings, especially considering the critiques provided. We have illustrated all of the above character states with photographs and CT data where relevant in the Supplemental Appendices.
- We have also integrated some sensitivity analyses to explore more traditional placements for *Colobops* among Diapsida, such as Sphenodontia. Although the hypothesis presented in the original submission remains the best supported of all, alternatives are only slightly less parsimonious. This owes largely to the limited amount of phylogenetic character data available for the specimen, but this uncertainty is now made explicit in several places in the manuscript.

"Beyond the phylogenetic placement of the new taxon, the authors found the enigmatic kuehneosaurids as deeply nested within Archosauromorpha, a result that strongly expands the ecomorphological disparity of the group during the Triassic. The relevance of this result competes with the importance of *Colobops* for the early evolutionary history of saurians. However, this result is not justified at all in the main text, neither in the supplementary information. I the position of kuehneosaurids as deeply nested within Archosauormorpha should be discussed and

clearly justified in this manuscript.”

- The recovery of Kuehneosauridae within Archosauromorpha is not entirely novel; it is consistent with a prior iteration of this analysis published in (Pritchard and Nesbitt, 2017) and prior hypotheses by (Evans, 1988). We concur with Reviewer 1 that this result is certainly worthy of note, but reviewing this result in detail will be the purview of a separate publication.

I consider that this manuscript is interesting and worth of publication, but the authors should successfully address the points raised above in the revised version of the manuscript. Thus, I recommend major changes for this manuscript.

Minor comments (main text)

“Lines 62–64: “Its presence at the dawn of the Mesozoic indicates that the major modern vertebrate clades originated in a world populated by small- and large-bodied ecomorphological extremes.”

I think that the relevance of Colobops for this statement is inflated. There are multiple small-bodied tetrapod taxa during the early Mesozoic that considerably increases the ecomorphological disparity of the clade, such as procolophonids and early

lepidosauromorphs. I would suggest rewording the sentence in a way that the presence of the new taxon increases or enriches the ecomorphological disparity, but not indicates by itself the presence of such ecomorphological extremes. “

- We agree with Reviewer 1 about the importance of these already-known small reptiles. We have changed the wording to suggest that *Colobops* emphasizes this pattern, rather than revealing an entirely new one.

Line 96: delete “, ”.

- Corrected.

Line 190: delete “small”.

- Corrected.

Line 210: “represented” is misspelled.

- Corrected.

Line 313: check if “mA” would be changed to “ μ A”.

- Corrected.

Line 328: delete repeated “was”.

- Corrected.

Line 336: “midpoint” is misspelled.

- Corrected.

Line 348, Phylogenetic analysis: can you add in this section how many characters and terminals has the modified version of the data matrix?

- Added to the methods section in the main text.

Line 353, Phylogenetic analysis: can you add how many trees have you recovered after the search and how many times the best score was recovered in the replicates?

- Added to the methods section in the main text.

Reference 55: “Azendohsaurus madagaskarensis” has to be in italics.

- Corrected.

Figure caption 1: indicate that the central bars of the postorbital on both sides are rendered in grey to show ambiguity in the reconstruction.

- Corrected.

*Figure caption 2: “Scaphonyx fischeri” is currently considered a nomen dubium (see Langer and Schultz, 2000). As a result, I suggest identify this specimen as *Hyperodapedon* sp.*

- We retained the name *Scaphonyx fischeri* for the OTU in this figure as that is the only name that has been used to refer to MCZ 1664 in the literature. If the editors would think it useful, we could also simply refer to the OTU by its specimen #.

*Figure caption 2: MCZ 1636 is referred by Sill (1970) to “Scaphonyx fischeri”. Thus, can you justify somewhere in the text why you are referring this specimen to *Hyperodapedon sanjuanensis*?*

- We restored the identification of MCZ 1636 to “Scaphonyx.”

Figure caption 4: “species-level” is misspelled. Figure 1: there are some white lines in the black background that should be deleted.

- Corrected.

REVIEWER 2 –

“

Reviewer #2 (Remarks to the Author):

This is an important announcement of a new Triassic reptile, and well written and illustrated. Its importance is that the new species is sister to the rhynchosaurs, a major clade of herbivores, and offers a very new insight into diversity and adaptation. The focus on a small, rather than, large animal is also of interest. This is a notably weird little critter, showing many rhynchosaur-like features (reinforced snout, unusually broad adductor chamber), and yet, as sister to Rhynchosauria, enforcing an unexpected 30-million-year long ghost range.

Methodologically, this is a nice paper because the rather tatty specimen has been CT scanned, so revealing important details inside the block which would otherwise not have been visible.

Can the authors be 100% confident about the great posterior width of the skull/ size of adductor chamber in that this is demarcated only on the left-hand side, and it depends entirely on the position chosen for the squamosal (?) – the purple bone with bifurcated anterior process – that is adrift from the other skull bones.”

- We consider the squamosal to be at or near its original position for the following reasons:
 - o I) The facet on the anterior process of the squamosal for the postorbital is positioned very near the preserved portion of the posterior process of the postorbital. Assuming the squamosal was re-positioned post-mortem would require a strong medial inclination of the postorbital posterior process that is not supported by the preserved fossil.
 - o II) None of the other bones or bone fragments that are weakly articulated or not articulated appear to be re-positioned.
 - All of the fragments of the nasal are positioned at or near the same depth in the matrix as the well-articulated bones of the roof of the rostrum.
 - The fragments of the pterygoid anterior processes are “wedged” between the two palatines despite a loose articulation. This is the expected position based on fossil and modern diapsid reptiles.

“192: saggittal = sagittal”

- Corrected.

“210: represneted = represented”

- Corrected.

REVIEWER 3 –

“

Reviewer #3 (Remarks to the Author):

This is a very interesting paper on a small reptile from the late Triassic Newark Supergroup. Small animals are really important because they often demonstrate evolutionary novelties, and are underrepresented in the fossil record.

The authors describe a small saurian with possible affinities to archosauromorphs. This small specimen seems to preserve dramatically expanded space for the jaw adductor musculature (although sadly the lower jaws are absent), which has implications for feeding adaptations of the animal. This is all really interesting and worth publishing.

However, it is the quantification for this claim that I think requires a little more work on the part of the authors. When something that is extremely small also has an “unusually large” anything, one should immediately suspect an allometric affect, as being small magnifies the relative size of structures. No study of allometry was presented by the authors, so the reader has no means for judging whether the jaw musculature architecture is absolutely larger or not.”

- We concur that some soft tissue structures, especially in the head, scale negatively relative to the overall size of both the head and the animal. However, across extant Lepidosauria and Archosauria, the jaw

adductors, their attachment sites on the skull roof, and the overall transverse breadth of the supratemporal fossae all scale positively with the overall size of the head and animal.

- To illustrate this scaling, a critical idea in this manuscript, we present a new Figure 4. This figure presents our data on the proportional breadth of the adductor chamber within a single ontogenetic trajectory: that of *Iguana iguana*. This animal undergoes major changes to its skull roof through ontogeny, eventually attaining a midline sagittal crest when the skull is approximately 3 centimeters wide. We also present references to show that this trend is true across modern Diapsida with supratemporal fenestrae.
- This scaling would only further emphasize the bizarre nature of the adductor chamber in *Colobops noviportensis*. In all of the modern lepidosaurs we sampled, including those with relatively large adductor chambers and prominent sagittal crests as adults, the attachment sites for the adductors are relatively much smaller when they are at a skull size similar to the *C. noviportensis* holotype. The relative development of the adductor chamber remains unique for an animal of this skull size, and the possibility of such hyper-development in a juvenile would be still more remarkable.

“The authors present their Figure 3 in which the proof for the large size is presented. Their argument is that their specimen is “a clear outlier”. I am uncertain that is clear--in fact it looks like it is within the cloud of the distributed data presented. Looking at some of the other bivariate plots presented in the supplemental information, individual species exhibit possible non linear trends (where sufficient data are known). Axes should therefore be log transformed. I suspect when this is done that outlier would be brought into the total range of variation--if not then its “outlierness” can be tested via residual analysis, or discriminant analysis (especially when size can be removed as a variable).”

- We concur that “outlier” has specific statistical properties that have not been substantiated among the available data. We have thus removed that term, and instead refer to the expansion of known morphospace represented by *Colobops noviportensis* among sampled diapsids.
- Although the morphospace plot presented in original submission Figure 3 was not labeled as such, the x-axis (representing the transverse breadth of the skull) is on a logarithmic scale. We have explicitly stated this on the labels for the axis in Figure 3 and all supplemental figures that incorporate morphospace plots with log-scaled axes. For the sake of completeness, I also incorporated a plot of the same dataset is not log-scaled (Figure S3 in the supplement).

“The rest of the paper is fine and very interesting. The figures are all good, although the palatal morphology is really small and hard to see clearly. I wish there was more braincase material! The phylogenetic analysis is based on a data set that is well vetted from the literature; characters all appear independent and the new characters well defined, as to be expected from this research group. There is a typo in Figure 4 caption. General support metrics for the presented strict consensus tree would be helpful to be included.”

- Typos have been corrected.
- Bremer support values and frequency differences from a Jackknife analysis are presented in the supplemental figures detailing the strict consensus.

Reviewers' comments:

Reviewer #1 (Remarks to the Author):

The authors did a good job addressing the changes proposed by the reviewers. I think that the manuscript is suitable for publication in the journal, but I think that the following further changes should be considered by the authors:

- 1) Figure 1 still has white lines between the different components of the figure.
- 2) The holotype and only known specimen of the new species is possibly based on a juvenile individual. As a result, despite of the result of the phylogenetic analysis, I suggest the authors to be more cautious through the text and the Systematic Palaeontology section at the time of considering it as a rhynchosaur.
- 3) Figure caption 2: *Scaphonyx fisheri* is not a valid rhynchosaur genus and species. As a result, this specimen should be quoted as "*Scaphonyx fisheri*" and it should be indicated here that it represents very likely a juvenile specimen sensu Benton and Kirkpatricki (1989).
- 4) Figure caption 2: this specimen was not referred by Sill (1970) to *Hyperodapedon sanjuanensis* but to "*Scaphonyx fisheri*". As a result, this specimen should be considered a referred specimen of "*Scaphonyx fisheri*" or an indeterminate hyperodapedontine rhynchosaur because of the absence of a taxonomic revision of *Hyperodapedon sanjuanensis*. It should be noted in the caption that this specimen likely represents a sub-adult or adult individual.

Reviewer #2 (Remarks to the Author):

No comments.

Reviewer #3 (Remarks to the Author):

I thank the authors for making the effort of accommodating my, and the other reviewers, comments on their previous draft. I am largely pleased with the results but I think their efforts highlighted a couple of remaining outstanding issues that should be addressed before publication.

I thank the authors for the new figures 3 and 4. This has now got me thinking about how they can best present what they want to prove--the greater expansion of the adductor chamber (AC). This seems to assume that the length of the AC remains the same, but this is not shown. Instead, the scatter plot has the proportion of width of AC regressed upon the log total width, which seems an odd choice (and why not log transform the Y axis as well? This would remove the curve seen in figure 4). Muscles performance is a function of the total area of the belly (setting aside the issue of pinnate or compound structure of muscle).

I wonder whether they might achieve their goal better by regressing a measure of the total area of the AC (say LxW) on the basal skull length (BSL), since BSL is strongly correlated to overall animal size (as demonstrated in multiple studies over the years. If the authors have a good reason for just focusing on width of the skull and width of the AC then that should be explicitly justified.

The new species still seems to fall out within the distribution of the overall plot in figure 3. At a minimum, it would be nice to see a 95% confidence interval placed for the overall population. Having the "blue star" (as mentioned in the figure caption but the new figure 3 has replaced this with an image of the skull so this requires amendment) if the authors remain against performing more formal statistical tests to show their specimen falls outside normal variation.

In the abstract, line 28: the use of "pan" here is unusual. A pan group is a stem group, but their use is that a "pan" is a taxon on a stem group. It would be more correctly an early diverging taxon on the stem, not an early diverging "pan".

Lines 192-210: this is all qualification, but with their data it should be relatively easy to quantify this.

Reviewers' comments: Reviewer #1 (Remarks to the Author):

The authors did a good job addressing the changes proposed by the reviewers. I think that the manuscript is suitable for publication in the journal, but I think that the following further changes should be considered by the authors:

1) *Figure 1 still has white lines between the different components of the figure.*

- We have checked the Adobe Illustrator export options and the current version of Figure 1 should not have these problems. We will provide original .pdf and .ai files to the journal.

2) *The holotype and only known specimen of the new species is possibly based on a juvenile individual. As a result, despite of the result of the phylogenetic analysis, I suggest the authors to be more cautious through the text and the Systematic Palaeontology section at the time of considering it as a rhynchosaur.*

- We concur with the Reviewer that the systematic affinities of *Colobops noviportensis* are open to question. We have added a few comments to a) highlight our analysis of alternative topologies detailed in the appendices and b) encourage further study of early saurian phylogeny.

3) *Figure caption 2: Scaphonyx fisheri is not a valid rhynchosaur genus and species. As a result, this specimen should be quoted as "Scaphonyx fisheri" and it should be indicated here that it represents very likely a juvenile specimen sensu Benton and Kirkpatricki (1989).*

- We have reworked all references to this specimen (Museum of Comparative Zoology 1664) as 'juvenile hyperodapedontine,' and now include a note in Supplemental Appendix C noting the confusions to its taxonomic history.

4) *Figure caption 2: this specimen was not referred by Sill (1970) to Hyperodapedon sanjuanensis but to "Scaphonyx fisheri". As a result, this specimen should be considered a referred specimen of "Scaphonyx fisheri" or an indeterminate hyperodapedontine rhynchosaur because of the absence of a taxonomic revision of Hyperodapedon sanjuanensis. It should be noted in the caption that this specimen likely represents a sub-adult or adult individual.*

- Apologies for not fixing this one last time. The operational taxonomic unit is now referred to as "juvenile hyperodapedontine (MCZ 1664)" on the phylogeny figure.

Reviewer #2 (Remarks to the Author): No comments.

Reviewer #3 (Remarks to the Author):

I thank the authors for making the effort of accommodating my, and the other reviewers, comments on their previous draft. I am largely pleased with the results but I think their efforts highlighted a couple of remaining outstanding issues that should be addressed before publication. I thank the authors for the new figures 3 and 4. This has now got me thinking about how they can best present what they want to prove--the greater expansion of the adductor chamber (AC). This seems to assume that the length of the AC remains the same, but this is not shown. Instead, the scatter plot has the proportion of width of AC regressed upon the log total width, which seems an odd choice (and why not log transform the Y axis as well? This would remove the curve seen in figure 4). Muscles performance is a function of the total area of the belly (setting aside the issue of pinnate or compound structure of muscle). I wonder whether they might achieve their goal better by regressing a measure of the total area of the AC (say $L \times W$) on the basal skull length (BSL), since BSL is strongly correlated to overall animal size (as demonstrated in multiple studies over the years. If the authors have a good reason for just focusing on width of the skull and width of the AC then that should be explicitly justified.

- Reviewer 3 brought up an interesting point that our use of transverse width of the supratemporal fossa is limited, although we emphasize that that metric correlates closely with the presence or absence of a midline sagittal crest: a hallmark of diapsid species with relatively massive adductor chambers. We present two graphs illustrating i) the absolute size of the transverse width of the adductor chamber (Figure 3A) and ii) the relative contribution of the adductor chamber to the transverse width of the skull (presented in the initial submissions and here in Figure 3B). Unfortunately, total lengths are available for a limited proportion of the sample in our dataset; even in some of the best-preserved fossils, the premaxilla and much of the anterior tips of the nasals are missing. To make use of the most specimens in our sample, we would prefer to use the transverse width measure for our x-axes.

- However, we think exploring a surface area metric to examine the proportions of the adductor chamber in Triassic Diapsida is an interesting avenue to pursue. For the subset of specimens in our study for which it was possible, we generated surface areas for the supratemporal fossae on the skull roof in ImageJ (Schindelin *et al.* 2012) and obtained anteroposterior lengths of the skulls. We use the length/width measurements for each skull to generate a proxy for skull dorsal surface

area. We present Figure 3C, which illustrates the log-transformed surface area metric against the proportional contribution of the supratemporal fossa area to the total area of the dorsum of the skull. Although there is absolutely no trend to speak of, the relative contribution of the supratemporal fossae to the skull roof in *Colobops* is far higher than any comparably sized animal in our analysis. This further supports the argument that the species represents an anatomical extreme relative to known, similar-sized diapsids.

The new species still seems to fall out within the distribution of the overall plot in figure 3. At a minimum, it would be nice to see a 95% confidence interval placed for the overall population. Having the "blue star" (as mentioned in the figure caption but the new figure 3 has replaced this with an image of the skull so this requires amendment) if the authors remain against performing more formal statistical tests to show their specimen falls outside normal variation.

- For the plots in which a clear trend between x and y variables is evident, we have calculated regressions between the two variables in R. The equations and r^2 values are presented for these two plots. In both cases, we present prediction intervals for the datasets not including *Colobops noviportensis*. *C. noviportensis* falls above of the 90% prediction interval for the plots of the log-transformed values of skull width against supratemporal fossa width. It falls above the 95% prediction interval for the plot of log-transformed skull width against the proportional breadth of the supratemporal fossae. The blue star is emphasized. We did not log-transform the proportion axes in our plots following recommendations from (Sokal & Rohlf 1995).

In the abstract, line 28: the use of "pan" here is unusual. A pan group is a stem group, but their use is that a "pan" is a taxon on a stem group. It would be more correctly an early diverging taxon on the stem, not an early diverging "pan".

- I believe this was a typographical error in our first post-review submission, which simply stated that our analyses "support *C. noviportensis* as an early-diverging pan." It should read "an early-diverging pan-archosaur," in line with the results of the phylogenetic analysis.

Lines 192-210: this is all qualification, but with their data it should be relatively easy to quantify this.

- Agreed. We have made the references for these values more explicit and provided a citation of our data table (Supplemental Table 1).

** See Nature Research's author and referees' website at

www.nature.com/authors for information about policies, services and author benefits

REFERENCES –

Schindelin, J., Arganda-Carreras, I., Frise, E., Kaynig, V., Longair, M., Pietzsch, T., Preibisch, S., Rueden, C., Saalfeld, S., Schmid, B. & others. 2012. Fiji: an open-source platform for biological-image analysis. *Nature methods*, 9, 676–682.

Sokal, R. F. & Rohlf, F. J. 1995. *Biometry*. 3rd ed. W. H. Freeman & Company, New York, 850 pp.

REVIEWERS' COMMENTS:

Reviewer #3 (Remarks to the Author):

I again thank the authors for so thoughtfully accommodating my comments from the prior draft. I have nothing further; this draft is acceptable in my opinion.